# Crinamine Induces Apoptosis and Inhibits Proliferation, Migration, and Angiogenesis in Cervical Cancer SiHa Cells

**DOI:** 10.3390/biom9090494

**Published:** 2019-09-16

**Authors:** Phattharachanok Khumkhrong, Kitiya Piboonprai, Waraluck Chaichompoo, Wittaya Pimtong, Mattaka Khongkow, Katawut Namdee, Angkana Jantimaporn, Deanpen Japrung, Udom Asawapirom, Apichart Suksamrarn, Tawin Iempridee

**Affiliations:** 1National Nanotechnology Center (NANOTEC), National Science and Technology Development Agency, Pathum Thani 12120, Thailand; phattharachanokkhumkhrong@gmail.com (P.K.); kitiya.tangmay@gmail.com (K.P.); wittaya.pimtong@nanotec.or.th (W.P.); Mattaka@nanotec.or.th (M.K.); katawut@nanotec.or.th (K.N.); angkana.jan@ncr.nstda.or.th (A.J.); deanpen@nanotec.or.th (D.J.); udom@nanotec.or.th (U.A.); 2Department of Chemistry and Center of Excellence for Innovation in Chemistry, Faculty of Science, Ramkhamhaeng University, Bangkok 10240, Thailand; waraluck_kik@hotmail.com (W.C.); asuksamrarn@yahoo.com (A.S.)

**Keywords:** crinamine, *Crinum asiaticum*, cervical cancer, tumor spheroid, zebrafish

## Abstract

*Crinum asiaticum* is a perennial herb widely distributed in many warmer regions, including Thailand, and is well-known for its medicinal and ornamental values. Crinum alkaloids contain numerous compounds, such as crinamine. Even though its mechanism of action is still unknown, crinamine was previously shown to possess anticancer activity. In this study, we demonstrate that crinamine was more cytotoxic to cervical cancer cells than normal cells. It also inhibited anchorage-independent tumor spheroid growth more effectively than existing chemotherapeutic drugs carboplatin and 5-fluorouracil or the CDK9 inhibitor FIT-039. Additionally, unlike cisplatin, crinamine induced apoptosis without promoting DNA double-strand breaks. It suppressed cervical cancer cell migration by inhibiting the expression of positive regulators of epithelial-mesenchymal transition SNAI1 and VIM. Importantly, crinamine also exerted anti-angiogenic activities by inhibiting secretion of VEGF-A protein in cervical cancer cells and blood vessel development in zebrafish embryos. Gene expression analysis revealed that its mechanism of action might be attributed, in part, to downregulation of cancer-related genes, such as AKT1, BCL2L1, CCND1, CDK4, PLK1, and RHOA. Our findings provide a first insight into crinamine’s anticancer activity, highlighting its potential use as an alternative bioactive compound for cervical cancer chemoprevention and therapy.

## 1. Introduction

Cervical cancer is the fourth most common female malignancy worldwide and contributes to approximately 530,000 new cases with 270,000 deaths annually [1]. It is still the second most common cancer and the leading cause of cancer mortality in Thai women, even though the nationwide incidence has decreased annually by 4.4% between 2000 and 2012 [2,3]. Current treatment can cure 60–90% of patients diagnosed at an early stage of the disease, yet prognosis for those with advanced or recurrent cervical cancer remains poor, with a five-year survival rate of 16.5% [4]. Chemotherapy is often used for treating advanced, recurrent, or metastatic disease; however, current regimens are hindered by limited response, cytotoxicity, and drug resistance [5,6,7]. Thus, more effective drugs with reduced side effects are urgently needed. This has brought attention to plants, which represent the direct or indirect source of over 60% of anticancer agents [6,8].

Amaryllidaceae is a large family of flowering plants comprising approximately 1100 species from 75 genera [9]. The family has drawn vast interest due to its unique alkaloids with antibacterial, antiviral, anticancer, and antimalarial activities [10]. Their medicinal properties have been further appreciated after the discovery of pancratistatin as a promising chemotherapeutic candidate, as well as the commercialization of galanthamine as an Alzheimer’s drug [11,12]. *Crinum asiaticum* (phlap-phlueng in Thai), a member of the Amaryllidaceae, is distributed in China, India, Sri Lanka, Malaysia, Japan, and Thailand [13]. It is generally cultivated as an ornamental plant and has a long history in traditional medicine throughout the region [14]. In Thailand, the plant species has recently been revised as *C. asiaticum* var. *asiaticum* [15]. *Crinum* alkaloids contain numerous compounds including lycorine and (+)-crinamine (hereinafter referred to as "crinamine"), and have been shown to possess important antitumor, antibacterial, and antifungal properties, as well as immuno-stimulating effects [16].

Crinamine was shown to exhibit cytotoxicity against a series of tumor cell lines including human oral epidermoid carcinoma (KB), colorectal cancer cells (COL-1), breast cancer cells (ZR-75-1), and glioblastoma (U-373) [17]. In addition, it was demonstrated to selectively induce apoptosis in rat hepatoma cells (5123tc) but not in non-cancerous human embryonic kidney cells (HEK-293T) [18]. Moreover, crinamine exerts anti-inflammatory activity by inhibiting nitric oxide induction in lipopolysaccharide-activated macrophages [19] and appears to inhibit HIF-1-induced gene transcription in a reporter assay system [20]. Although a mechanism of how crinamine affects cancer cells remains largely unknown, accumulating evidence suggests it may be a promising anticancer agent.

In this study, we purified crinamine from bulbs of *C. asiaticum* var. *asiaticum* and investigated its cancer-specific cytotoxicity on a cervical cancer cell lines relative to normal cells. We further evaluated its potency in inhibiting anchorage-independent growth of tumor spheroids with respect to commonly used chemotherapeutic drugs and CDK9 inhibitors. Additionally, we tested crinamine’s effect on apoptosis, double-strand DNA damage, and cancer cell migration, as well as *in vivo* angiogenesis in zebrafish embryos. Finally, we explored potential downstream target genes of crinamine by profiling the expression of cancer-related genes in cervical cancer cells. This study provides a first report on the anti-cervical cancer activity of crinamine and highlights its potential as an alternative compound for chemoprevention or cancer therapeutics.

## 2. Materials and Methods

### 2.1. Extraction and Isolation

The bulbs of *C. asiaticum* var. *asiaticum* were collected in Nonthaburi Province, Thailand. A voucher specimen is deposited at the Faculty of Science, Ramkhamhaeng University, Thailand (Apichart Suksamrarn, No. 083). The minced and fresh bulb (200 g) was extracted with methanol (MeOH) (5 × 5 L) following incubation for three days at room temperature with frequent stirring. The solution was filtered and the solvent was evaporated to yield the crude extract (0.65 g). The extract was resuspended in 250 mL distilled water and sequentially partitioned with n-hexane (3 × 1 L), ethyl acetate (EtOAc) (3 × 1 L), and n-butanol (3 × 1 L) to yield hexane (22 mg), EtOAc (155 mg), and butanol (250 mg) extracts, respectively. The EtOAc extract was then fractionated by column chromatography over silica gel 60 (0.063–0.200 mm; Merck, Darmstadt, Germany), using a gradient of EtOAc, EtOAc-MeOH, and MeOH, respectively, to give E1 (15 mg) and E2 (82 mg) fractions. The E2 fraction was further separated into seven fractions (E3–E9) by column chromatography under isocratic elution conditions (10% MeOH in CH_2_Cl_2_). Fraction E8 (13 mg) was then separated on a Sephadex LH-20 column (GE Healtcare, Uppsala, Sweden), eluted with MeOH, followed by silica column chromatography and elution with 5–15% MeOH in EtOAc to yield 6 mg of crinamine. Optical rotations were measured on a JASCO-1020 polarimeter (Jasco, Tokyo, Japan). Infrared (IR) spectra were obtained using a Frontier Fourier transform infrared (FT-IR) spectrometer (Perkin-Elmer, Norwalk, CT, USA). ^1^H and ^13^C nuclear magnetic resonance (NMR) spectra were recorded on an AVANCE 400 FT-NMR spectrometer (Bruker, Billerica, MA, USA) operated at 400 MHz (^1^H) and 100 MHz (^13^C). Electrospray ionization time-of-flight mass spectrometry (ESI-TOF-MS) spectra were acquired with a Bruker micrOTOF mass spectrometer (Bruker). The spectroscopic (NMR and mass spectra) data were consistent with those reported previously [17].

### 2.2. Cell Culture

Human primary dermal fibroblasts (HDFs), adult HDFa (PCS-201-012), immortalized human papilloma virus (HPV)-16 E6/E7 transformed ectocervical Ect1/E6E7 (CRL-2614), HPV-positive cervical carcinoma (HeLa (CCL2) and SiHa (HTB-35)), and HPV-negative cervical carcinoma C33a (HTB-31) cell lines were purchased from the American Type Culture Collection (ATCC, Manassas, VA, USA). The immortalized human keratinocyte (HaCaT) cell line was a gift from Arthit Chairoungdua, Mahidol University, and its identity was confirmed by STR profiling performed by the ATCC. HDFa cells were cultured in Fibroblast Basal Medium (PCS-201-030, ATCC) supplemented with Fibroblast Growth Kit—Low-serum (PCS-201-041, ATCC). Ect1/E6E7 cells were grown in keratinocyte serum-free medium (SFM) supplemented with Keratinocyte-SFM Supplement (Gibco, Thermo Fisher Scientific, Waltham, MA, USA). HeLa, SiHa, C33a, and HaCaT cell lines were cultured in Dulbecco’s Modified Eagle Medium (DMEM; HyClone, Thermo Fisher Scientific) including 10% fetal bovine serum (FBS; HyClone). All cells were maintained at 37 °C in a humidified incubator with 5% CO_2_. Cell lines below passage number 21 and primary fibroblasts below passage number 6 were used in all experiments.

### 2.3. Cell Viability Assay

Cells were seeded in 96-well plates and grown in culture medium. They were treated with crinamine or 0.5% dimethyl sulfoxide (DMSO) at the indicated concentration for 48 h upon reaching 80% confluence. Cell viability was measured using the CellTiter 96^®^ AQueous Non-Radioactive Cell Proliferation Assay (MTS) (Promega, Madison, WI, USA) and normalized to the DMSO control.

### 2.4. Cell Proliferation Assay Using Real-time Cell Analysis (RTCA)

The rate of cell proliferation was assessed in real-time using an xCELLigence RTCA dual-plate instrument (ACEA Biosciences, San Diego, CA, USA) as previously described [21]. Briefly, HaCaT (8000 cells/well) and SiHa (6000 cells/well) were seeded into 100 µL of DMEM with 10% FBS in E-plates (ACEA Biosciences). The impedance value, expressed as cell index, was recorded every 30 min. Cells were treated with 50 µL of the indicated concentration of crinamine or 0.5% DMSO in growth medium when they reached log phase and impedance was recorded for another 96 h. The cell proliferation rate was derived from the slope of the line between two given time points.

### 2.5. Multicellular Tumor Spheroid Assay

SiHa and C33a cells were seeded at a density of 4000 cells/well in ultra-low attachment 96-well plates (Corning, Thermo Fisher Scientific) and incubated in DMEM with 10% FBS for seven days with 50% medium replacement every 2–3 days. Subsequently, tumor spheroids were incubated with the indicated concentration of BAY-1143572 (Atuveciclib) (MedChemExpress, Monmouth Junction, NJ, USA), crinamine, cisplatin (BioVision, Milpitas, CA, USA), FIT-039 (Aobious, Gloucester, MA, USA), carboplatin (Sigma-Aldrich, St. Louis, MI, USA), 5-fluorouracil (5-FU) (Sigma-Aldrich), gemcitabine (Sigma-Aldrich), or 0.5% DMSO for 48 h. Cell images were taken using an Is71 microscope (Olympus, Tokyo, Japan) and cell viability was determined using the CellTiter-Glo^®^ 3D cell viability assay (Promega).

### 2.6. Annexin V/Propidium iodide (PI) Apoptosis Assay

SiHa cells were seeded in 24-well plates and treated with the indicated concentration of cisplatin, crinamine, or 0.16% DMSO for 16 h. Cells were stained with Annexin V-fluorescein isothiocyanate (FITC) using the Annexin V-FITC Early Apoptosis Detection Kit (#6592; Cell Signaling Technology, Danvers, MA, USA). Images of Annexin V-FITC-positive or PI-positive cells were taken using an Is71 microscope. Fluorescence intensity of Annexin V-FITC-positive cells was quantified at excitation/emission wavelengths of 495 nm/525 nm using a Synergy^TM^ H1 Multi-Mode Reader (BioTek, Winooski, VT, USA) and normalized to the DMSO control. Assays were performed in triplicate.

### 2.7. Caspase-3/7 Activity Assay

SiHa cells were seeded in 96-well plates and grown for 24 h. Cells were treated with cisplatin, crinamine, or 0.32% DMSO at the indicated concentration for another 24 h. Caspase-3/7 activity was measured using the Caspase-Glo^®^ 3/7 assay (Promega) in a Synergy^TM^ H1 Multi-Mode Reader and normalized to the DMSO control. Assays were performed in triplicate.

### 2.8. γ-H2AX Immunofluorescence Staining

SiHa cells were seeded in Nunc™ glass bottom dishes (Thermo Scientific), incubated overnight, and treated with cisplatin, crinamine, or 0.16% DMSO at the indicated concentration for 4 or 24 h. Cells were washed with cold phosphate-buffered saline (PBS), fixed with 4% paraformaldehyde, and permeabilized with 0.1% TritonX-100 and 0.1% bovine serum albumin in PBS. After washing, cells were blocked with 10% normal donkey serum in PBS, followed by incubation with rabbit anti-phospho-histone H2AX (Ser139) (#9718; Cell Signaling Technology) at 4 °C overnight. Subsequently, cells were washed with PBS, incubated with goat anti-rabbit IgG, DyLight^®^ 594 conjugated (ab96901; Abcam, Cambridge, UK), washed with PBS, and counterstained with DAPI (#4083; Cell Signaling Technology). Cell images were taken with a FV10i confocal laser scanning microscope (Olympus). Assays were performed in duplicate on two separate occasions.

### 2.9. Real-Time Cell Migration Assay

Real-time cell migration was measured using the xCELLigence RTCA system as previously reported [21]. Briefly, 175 µL of DMEM with 4% FBS and the indicated concentration of crinamine or 0.08% DMSO was added to the lower chamber of the CIM-16 plate (ACEA Biosciences). Then, 50 µL of serum-free medium (SFM) was added to the upper chamber, followed by background reading. Finally, crinamine or DMSO in 50 µL SFM was added to the upper chamber and SiHa cells (40,000 cells) in 50 µL SFM were added to all wells. SFM alone was used as negative control for both the lower and upper chambers. Impedance was read every 15 min for 30 h. Assays were performed in triplicate on two independent occasions.

### 2.10. Gene Expression Analysis by Quantitative Reverse Transcription PCR (RT-qPCR)

SiHa cells were seeded in 96-well plates and cultured for overnight. Cells were treated with crinamine or 0.12% DMSO for 8 h. Total RNA was isolated from cells using the Monarch^®^ Total RNA Miniprep kit (New England Biolabs, Ipswich, MA, USA) and quantified by a NanoDrop One UV-Vis spectrophotometer (Thermo Scientific). All samples exhibited optical density A260/A280 and A260/A230 ratios of ~2.0. RNA was reverse-transcribed to cDNA using the iScript^TM^ cDNA synthesis kit (Bio-Rad, Hercules, CA, USA). The qPCR reaction was performed in a Bio-Rad CFX96 Touch using a SsoAdvanced™ Universal SYBR Green Supermix (Bio-Rad). PCR cycles were as follows: 98 °C for 30 s, 40 cycles of 98 °C for 5 s, and 60–62.9 °C for 30 s, followed by melt curve analysis, from 65–95°C with 0.5°C increment per cycle. A list of primers, nucleotide sequences (or Bio-Rad assay IDs), and annealing temperatures (T_a_) are provided in Appendix A. Gene expression was normalized to at least two reference genes (U6 snRNA, RPS13, or PGK1) giving an average expression stability (M-value) <0.5, as determined by the GeNorm algorithm [22]. Assays were performed in duplicate or triplicate and analyzed by CFX Manager™ Software (Bio-Rad).

### 2.11. Quantification of Vascular Endothelial Growth Factor-A (VEGF-A) Secretion

SiHa cells were grown in 96-well plates and were treated with crinamine or 0.16% DMSO for two days. Expression of VEGF-A was quantified from cell culture medium using the Ella automated immunoassay platform (ProteinSimple, San Jose, CA, USA) and an analyte panel (R and D Systems, Minneapolis, MN, USA) as described earlier [23]. Briefly, 50 µL of two-fold-diluted culture medium and 350 µL of wash buffer were added to the cartridge, followed by placement into the instrument. The cartridge contained a built-in lot-specific standard curve and samples were run as triplicates by automatic splitting into three glass nano reactors. Data were analyzed using SimplePlex Explorer software (ProteinSimple).

### 2.12. Zebrafish Embryo Angiogenesis Assay

Zebrafish (*Danio rerio*) were maintained in a recirculating zebrafish housing system (AAB-074; Yakos65, Taipei, Taiwan) at 28.5 °C under a 14 h light/10 h dark photoperiod. The facility was approved by the National Research Council of Thailand. All procedures were approved by the Institutional Animal Care and Use Committees of the National Science and Technology Development Agency (no. 002-2560). To evaluate the effect of crinamine on blood vessel formation, embryos at 4 h post-fertilization were incubated with 0.08% DMSO or crinamine at the indicated concentration, with solution replacement every 24 h (12 embryos per treatment). After three days of incubation, subintestinal vessels (SIVs) were assayed by whole-mount alkaline phosphatase staining as described previously [24]. Briefly, embryos were fixed with 4% paraformaldehyde, soaked in methanol at −20 °C overnight, and permeabilized in ice-cold acetone, followed by washing with PBS containing Tween 20. After incubating with staining buffer (100 mM Tris-HCl (pH 9.5), 50 mM MgCl_2_, 100 mM NaCl, and 0.1% Tween-20), the reaction was developed by adding nitro blue tetrazolium/5-bromo-4-chloro-3-indolyl phosphate (Roche Diagnostics, Rotkreuz, Switzerland). The stained embryos were then photographed with an Olympus SZX16 stereomicroscope equipped with an Olympus DP73 camera and cellSens Standard software (Olympus). The number of SIVs in each embryo was manually counted and their area was quantified by ImageJ (NIH, Bethesda, MD, USA).

### 2.13. Statistical Analysis

Half maximal inhibitory concentration (IC_50_) values were determined by GraphPad Prism 8 (GraphPad Software, San Diego, CA, USA) using a dose-response plot [log(inhibitor) vs. response-variable slope]. Statistical analysis was performed using GraphPad InStat version 3 (GraphPad Software). Unpaired Student’s t-test and one-way ANOVA with post-hoc Tukey honestly significant difference were used for comparing two or more groups, respectively. *p* < 0.05 was considered statistically significant.

## 3. Results and Discussion

### 3.1. Crinamine Is More Cytotoxic to Cervical Cancer Cells Compared to Normal Cells

Crinamine was purified by column chromatography from the EtOAC fraction of the *C. asiaticum* var. *asiaticum* bulb extract. The chemical structure was identified by NMR (Figure 1a) and mass spectral and spectroscopic data were in agreement with that reported in a previous study (Appendix B) [17]. Its yield from the crude extract was estimated to be 0.92%. To screen for potential anticancer activity, we first investigated the cytotoxicity of crinamine in normal and cervical cancer cell lines using the MTS assay. As shown in Figure 1b, crinamine was significantly more cytotoxic to HPV-positive cervical carcinoma SiHa (IC_50_ = 23.52 µM) and HPV-negative cervical carcinoma C33a (IC_50_ = 60.89 µM) cells than primary fibroblasts HDFa (IC_50_ > 100 µM), ectocervical epithelial cell line Ect1/E6E7 (IC_50_ > 100 µM), and keratinocyte cell line HaCaT (IC_50_ = 85.45 µM). Second, compound selectivity was determined by a selectivity index (SI) expressed as a ratio of the IC_50_ of normal *vs*. neoplastic cells [25]. Relative to physiological relevant Ect1/E6E7cells (IC_50_ = 149.25 ± 16.05 µM), the SI for crinamine was 6.35 in SiHa and 2.45 in C33a cells. The high SI value (>2) indicates selective toxicity of crinamine against cervical cancer cells.

To confirm its specificity towards cancerous cells, we determined the effect of crinamine on the rate of cell proliferation in HaCaT and SiHa cells using the RTCA platform. Cells were treated with various concentrations of crinamine or DMSO, followed by quantification of the cell index every 30 min during four days. As shown in Figure 1c,e, increasing concentrations of crinamine inhibited cell proliferation in a dose-dependent manner in both cell lines. Notably, at 50 µM, crinamine completely inhibited the proliferation rate of SiHa cells but not that of HaCaT cells (Figure 1d vs. Figure 1f), supporting the increased toxicity against cervical cancer cells.

Moreover, crinamine has been documented to exert strong cytotoxicity against other cancer cell lines giving IC_50_ values ranging from 3.3–22.9 μM, including lung cancer (LUC-1), oral epidermoid carcinoma (KB and KB-V1), skin carcinoma (A-431), colorectal cancer (COL-1), breast cancer (BCA-1 and ZR-75-1), glioblastoma (U-373), and melanoma (MEL-2) [17]. A superior cytotoxicity of crinamine towards a broad range of cancer cell lines makes it an appealing bioactive compound and warrants further characterization.

### 3.2. Crinamine Inhibits Tumor Spheroid Growth with Higher Potency than Existing Chemotherapeutic Drugs

Next, we used the multicellular tumor spheroid model (MCTS) to compare the potency of crinamine and existing chemotherapeutic drugs in inhibiting anchorage-independent growth of cervical cancer SiHa and C33a cells. The model allows cells to grow in three dimensions, displaying a mixture of proliferating cells in the outer layer and quiescent cells in the inner layer [26]. Importantly, akin to the situation *in vivo*, cancer cells grown as spheroids can acquire resistance to many chemotherapeutic agents as a result of interaction with neighboring cells and their environment. Thus, the model can predict drug response more precisely than a monolayer cell culture [27,28].

To generate tumor spheroids, SiHa and C33a cells were cultured in an ultra-low attachment plate for 7 days, followed by treatment for two days with various concentrations of chemotherapeutic drugs (cisplatin, carboplatin, 5-FU, and gemcitabine), CDK9 inhibitors (BAY-1143572 and FIT-039), crinamine, or DMSO. The number of viable cells was measured and the IC_50_ value of each compound was determined. As shown in Figure 2, crinamine strongly inhibited tumor spheroid growth, producing IC_50_ values of 18.83 and 42.61 µM in SiHa and C33a tumor spheroids, respectively. The rank order of the compounds in inhibiting growth of SiHa spheroids was BAY-1143572 > crinamine > cisplatin > FIT-039 > carboplatin > 5-FU > and gemcitabine; while the rank order in C33a spheroids was BAY-1143572 > cisplatin ~ gemcitabine > crinamine ~ carboplatin > FIT-039 > 5-FU. Therefore, our data indicate that crinamine inhibits cervical cancer spheroid growth more efficiently than carboplatin, FIT-039, or 5-FU in both SiHa and C33a cells.

It is noteworthy that the highly selective CDK9/CycT1 inhibitor Atuveciclib (BAY-1143572) was the most potent drug in inhibiting tumor spheroid growth in both SiHa and C33a cells, yielding IC_50_ values of approximately 7 µM. A previous report similarly showed that BAY-1143572 exhibited very strong anti-proliferative activity against cervical cancer HeLa cells, with an IC_50_ value of 920 nM [29]. Another less potent CDK9/CycT1 inhibitor, FIT-039, was two-fold less effective than crinamine in targeting tumor spheroids, as manifested by an IC_50_ of 40 and 97 µM in SiHa and C33a cells, respectively. FIT-039 was found to hinder HPV replication and expression of E6 and E7 oncogenes in HPV^+^ cervical cancer cells, thereby restoring tumor suppressors p53 and pRb [30]. Whether BAY-1143572 acts through a similar mechanism as FIT-039 requires further investigation; nevertheless, our results indicate that BAY-1143572 is a very potent antineoplastic agent, with potential for clinical study of its effectiveness against cervical cancer.

### 3.3. Crinamine Induces Apoptosis in Cervical Cancer SiHa Cells without Activating DNA Damage

Next, we explored whether crinamine could induce apoptosis in cervical cancer cells by measuring phosphatidyl serine externalization using Annexin V staining. As shown in Figure 3a,b, treatment with 8 and 16 µM crinamine increased overall fluorescence intensity and the proportion of Annexin V-positive SiHa cells to a level comparable with 20 µM cisplatin (positive control) treatment, indicating activation of apoptosis. Additionally, we assessed the combined activities of caspase-3 and -7 as a measure of the caspase-dependent apoptotic pathway. Whereas treatment of SiHa cells with 10 and 20 µM cisplatin for 24 h led to a significant increase in caspase-3/7 activity, treating cells with crinamine did not. These findings suggest that crinamine promotes caspase-3/7-independent apoptotic cell death in cervical cancer cells (Figure 3c). It should be noted that we observed a decrease in caspase 3/7 activity following crinamine treatment. Although we are unable to explain this observation, we propose that this might be caused by non-specific degradation during a late stage of apoptosis or secondary necrosis. Alternatively, crinamine might negatively regulate caspase 3/7 activity at the later time point by regulating an expression of a caspase inhibitor, such as X-linked inhibitor of apoptosis protein (XIAP). Further experimental validation is required to provide a definitive answer.

Bcl-2 (BCL2) is a membrane protein that blocks apoptotic cell death. Its overexpression is evident in premalignant and malignant lesions of the cervix and has been suggested to play a vital role in the early stages of cervical tumorigenesis [31,32]. In addition, B-cell lymphoma xL (Bcl-xL) (BCL2L1) is an anti-apoptotic protein overexpressed in various tumors including cervical cancer [33]. To further examine the role of crinamine in apoptosis, we measured its effect on the mRNA expression of BCL2 and BCL2L1 by RT-qPCR. As shown in Figure 6, which reports several gene profiles, treatment of SiHa cells with 12 µM crinamine for 8 h significantly downregulated expression of BCL2L1 but not that of BCL2. Therefore, we conclude that crinamine induces apoptosis, in part, by inhibiting expression of the anti-apoptotic BCL2L1 gene.

Cancer chemotherapeutic drugs and radiotherapy kill cancer cells primarily by inducing DNA double-strand breaks (DSBs) [34]. Hence, we asked whether crinamine could induce DNA damage by measuring phosphorylation of H2AX at Ser 139 (γ-H2AX) using immunofluorescence. As evidenced in Figure 3d,e, incubating SiHa cells with 20 µM cisplatin for 4 and 24 h markedly increased phosphorylation of H2AX. In contrast, treatment with 8 and 16 µM crinamine did not induce γ-H2AX foci compared to the control. Therefore, we conclude that crinamine activates apoptotic cell death in SiHa cells without inducing DSBs.

Cisplatin has long been considered a backbone drug for treating cervical cancer as well as other solid tumors [35]. It primarily targets genomic DNA, causing a plethora of DNA lesions that block transcription and replication, eventually leading to cell death [36]. Despite its high efficacy, drug resistance and systemic toxicity arising from non-selective DNA damage remain a major challenge, limiting its clinical usefulness [37]. Thus, a bioactive compound that does not depend on DNA damage activity for its anticancer effect might represent a promising alternative. Pancratistatin, an Amaryllidaceae alkaloid isolated from *Hymenocallis littoralis*, is a prime example of such a compound. It is highly selective and promotes caspase-independent apoptosis in colon cancer cells without inducing DNA damage as a primary mechanism [38]. In addition, we have recently reported arborinine from *Glycosmis parva* as another example exerting its anti-cervical cancer activity without compromising DNA integrity [21]. Our discovery of crinamine as another compound with this unique property suggests that it might potentially serve as a better alternative to cisplatin for treating cervical cancer.

### 3.4. Crinamine Inhibits SiHa Cells Migration by Downregulating Expression of SNAI1 and VIM

A majority of cancer-related deaths are caused by metastasis. For cervical cancer, the five-year survival rate for metastatic disease is 16.5% compared to 91.5% for localized cancer, and there is no standard treatment due to its heterogeneous manifestations [39,40]. Because cell migration is an indispensable process required at every step of the metastatic cascade [41], we next explored the effect of crinamine on cervical cancer cell migration using a real-time cell migration assay based on the xCELLigence system. As shown in Figure 4a, SiHa cells treated with DMSO migrated from the upper chamber in SFM to the lower chamber containing 4% FBS, while no migration was observed in the SFM control. Treatment of SiHa cells with 2, 4, and 8 µM crinamine led to a significant decrease in cell migration in a dose-dependent manner (Figure 4a,b). Thus, our data prove that crinamine could inhibit cervical cancer cell migration.

The epithelial-mesenchymal transition (EMT) enables cancer cells to acquire migratory and invasive capabilities and plays a pivotal role in cervical cancer progression and metastasis [42,43]. EMT is regulated by various key transcription factors, including the zinc-finger SNAIL family, the zinc-finger E-box binding homeobox (ZEB) family, and the TWIST family of basic helix-loop-helix (bHLH) transcription factors [44]. Snail (SNAI1), ZEB1, ZEB2, and Twist1 have been shown to inhibit E-cadherin expression and induce the EMT process [45,46]. Vimentin (VIM) is an intermediate filament protein that functions during cell migration, serving as a biomarker and driver of EMT, as well as stimulating metastatic progression [47,48]. In addition, matrix metalloproteinase-1 (MMP1), induced following the EMT, promotes extracellular matrix degradation and contributes to tumor invasion and metastasis [49].

To understand how crinamine affected cancer cell migration, we determined whether crinamine inhibited expression of key genes involved in the EMT and metastasis by RT-qPCR. As shown in Figure 4c, treating SiHa cells with 12 µM crinamine for 8 h significantly inhibited expressions of SNAI1 and VIM, without affecting the levels of TWIST1, ZEB1, and ZEB2. Based on these findings, we conclude that crinamine inhibits cervical cancer cell motility, at least in part, by downregulating expression of key positive regulators of the EMT pathway. Determining whether crinamine could suppress *in vivo* metastasis will require further studies in animal models.

### 3.5. Crinamine Inhibits VEGF-A Expression and In Vivo Angiogenesis in Zebrafish Embryos

Angiogenesis is required to sustain solid tumor growth beyond 1–2 mm^3^ and is essential for tumor invasion and metastasis [50]. VEGF-A is one of the primary factors driving expansion of tumor vasculature [51]. Here, we tested whether crinamine affected angiogenesis by measuring the level of secreted VEGF-A using an automated immunoassay platform. As shown in Figure 5a, treating SiHa cells with 4, 8, and 16 µM crinamine for two days significantly decreased the secretion of VEGF-A into culture medium. The effect of crinamine on expression of VEGF-A transcript was assessed by RT-qPCR and, as reported in Figure 6, treating SiHa cells with 12 µM crinamine for 8 h significantly declined VEGF-A mRNA expression. Thus, our data indicate that crinamine possesses anti-angiogenic activity by inhibiting expression of VEGF-A at both mRNA and protein levels.

Next, we evaluated the anti-angiogenic property of crinamine *in vivo* by examining its ability to inhibit blood vessel development in zebrafish embryos. Treating the embryos with 4, 8, and 16 µM crinamine for three days did not promote any abnormal morphology, confirming that the concentrations used in the experiment were not toxic to the embryos. We then performed whole-mount alkaline phosphatase staining to visualize SIVs on the yolk of the embryos, followed by quantitative analysis of the number and area of SIVs (Figure 5b). As shown in Figure 5b–d, incubating zebrafish embryos with 16 µM crinamine for three days significantly reduced both the number and area of SIVs. We, therefore, conclude that crinamine could suppress *in vivo* angiogenesis in the zebrafish model.

Targeting angiogenesis has been proved effective in treating advanced cervical cancer. Addition of bevacizumab, an anti-VEGF monoclonal antibody, to paclitaxel-topotecan or paclitaxel-cisplatin was found to significantly improve overall survival in patients with persistent, recurrent, or metastatic cervical cancer [52,53]. Given that crinamine presents anti-angiogenic activity, we speculate that crinamine might be useful for treating advanced cervical cancer when combined with standard chemotherapeutic regimens. This hypothesis definitely requires further *in vivo* confirmation.

### 3.6. Crinamine Inhibits mRNA Expression of AKT1, BCL2L1, CCND1, CDK4, PLK1, RHOA, and VEGF-A in SiHa Cells

Finally, we attempted to elucidate the mechanism underlying crinamine’s negative effect on cervical cancer cells. To this end, we profiled the expression of 20 cervical cancer-related genes using RT-qPCR, including aurora kinase A (AURKA), aurora kinase B (AURKB), serine-threonine protein kinase AKT1 (AKT1), B-cell lymphoma 2 (BCL2), Bcl-2-like 1 (BCL2L1), baculoviral inhibitor of apoptosis repeat-containing 5 (BIRC5), mitotic checkpoint serine/threonine-protein kinase BUB1 beta (BUB1B), cyclin A2 (CCNA2), cyclin B1 (CCNB1), cyclin D1 (CCND1), cell-division cycle protein 20 (CDC20), cyclin-dependent kinase 4 (CDK4), epidermal growth factor receptor (EGFR), maternal embryonic leucine zipper kinase (MELK), PDZ binding kinase (PBK), polo-like kinase 1 (PLK1), pituitary tumor-transforming gene 1 (PTTG1), ras homolog gene family, member A (RhoA), DNA topoisomerase II alpha (TOP2A), and vascular endothelial growth factor A (VEGF-A).

As shown in Figure 6, treating SiHa cells with 12 µM crinamine for 8 h led to a significant decrease in mRNA expression of AKT1, BCL2L1, CCND1, CDK4, PLK1, RHOA, and VEGF-A; while having no effect on AURKA, AURKB, BCL2, BIRC5, BUB1B, CCNA2, CCNB1, CDC20, EGFR, MELK, PBK, PTTG1, and TOP2A. AKT1 is a key mediator in the phosphoinositide 3-kinase (PI3K)/AKT signaling cascade, regulating cell proliferation, cell cycle, and apoptosis, including tumorigenesis [54]. Hyperphosphorylation of AKT1 has been reported in cervical neoplastic tissues, suggesting constitutive activation of the PI3K/AKT pathway during cervical cancer development [55]. Further, cyclin D, interacting with CDK4 or CDK6, promotes the G1-to-S phase transition, leading to cell proliferation. Gene amplification or overexpression of cyclin D1 or CDK4 has been documented in cervical cancer, suggesting that targeting these factors could represent a possible strategy against cervical cancer [56,57].

Moreover, PLK1 is a serine-threonine kinase driving mitotic progression and is highly expressed in multiple cancers including cervical cancer. It is a downstream effector of the nonreceptor tyrosine kinase c-ABL and targeting both proteins could inhibit cervical cancer growth *in vivo* [58]. Lastly, Rho A is a conserved small GTPase involved in several cellular responses, including cytoskeleton organization, cell division, and transcription regulation. RhoA is overexpressed in cervical cancer and is associated with high metastatic potential [59]. Our data from Figure 4c and Figure 6 suggest that multiple anticancer activities of crinamine might be, in part, mediated by downregulation of these key oncogenic factors, leading to inhibition of cell proliferation, migration, angiogenesis, as well as stimulating apoptotic cell death.

## 4. Conclusions

In summary, we report that crinamine isolated from bulbs of *C. asiaticum* var. *asiaticum* exerts potent anticancer activity in cervical cancer cells. Its cytotoxicity is selective to cervical cancer cell lines and is more effective at inhibiting anchorage-independent tumor spheroid growth than several known chemotherapeutic drugs. Additionally, the compound activates DNA damage-independent apoptosis, reduces cell migration activity by downregulating key EMT inducers, and suppresses VEGF-A secretion and *in vivo* angiogenesis. On the basis of this study, we suggest that crinamine might be applicable for cancer chemoprevention or serve as an alternative agent for cervical cancer therapy.

## Figures and Tables

**Figure 1 biomolecules-09-00494-f001:**
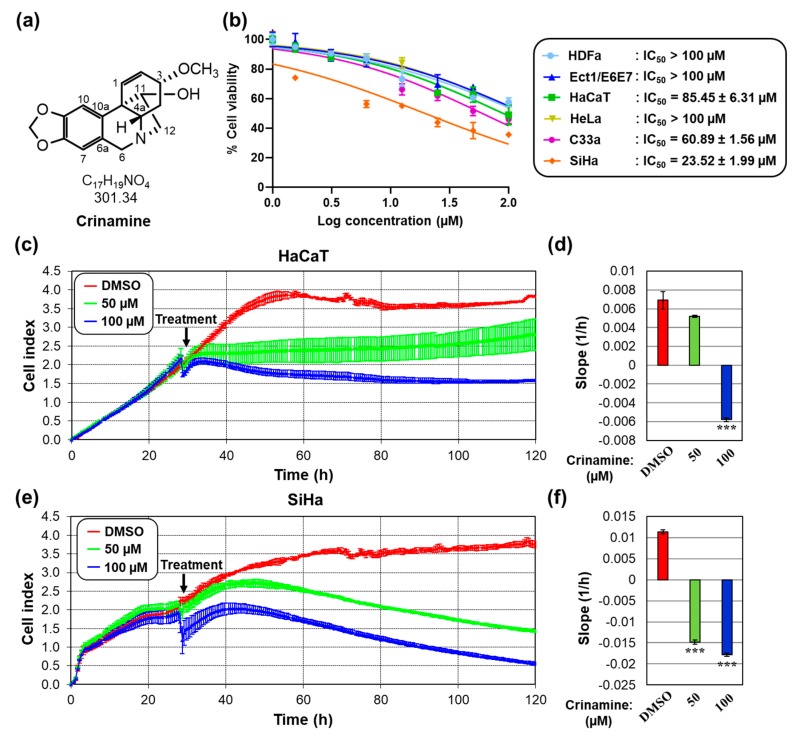
Selective cytotoxicity of crinamine against cervical cancer cell lines. (**a**) Structure of crinamine isolated from *C. asiaticum* var. *asiaticum* as determined by spectroscopic data. (**b**) Inhibitory dose-response curves of HDFa, Ect1/E6E7, HaCaT, HeLa, C33a, and SiHa cells treated for two days with crinamine. Cell viability was determined and normalized to DMSO. IC_50_ values were calculated from triplicates obtained from two independent experiments and are represented as the mean ± standard deviation (SD). (**c**–**f**) Real-time proliferation of (**c,d**) HaCaT and (**e,f**) SiHa cells following treatment with crinamine. Cells grown in E-plates were incubated with DMSO or the indicated concentrations of crinamine when the cell index was approximately 2.0. (**c,e**) The rates of cell proliferation were continuously monitored for four days after treatment. (**d,f**) The slope was calculated between 30 and 120 h time points and is reported as mean ± SD from duplicate measurements. Results presented here are typical of data observed on two separate occasions. ****p* < 0.001 *vs.* control.

**Figure 2 biomolecules-09-00494-f002:**
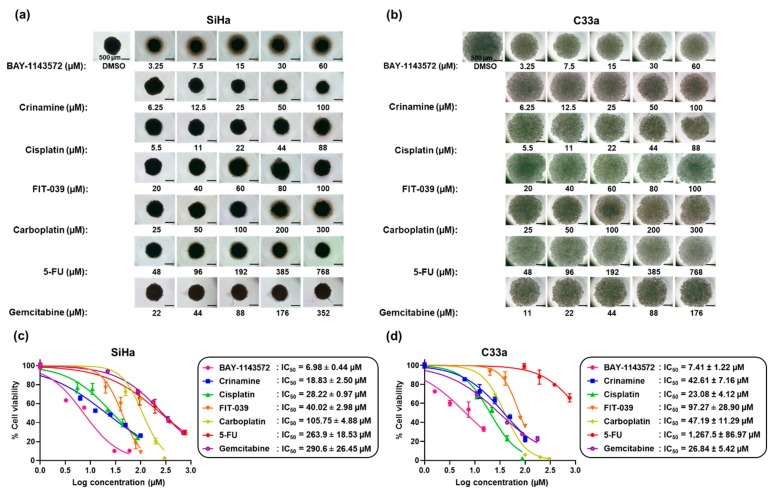
Comparative cytotoxicity of CDK9 inhibitors, chemotherapeutic drugs, and crinamine in a three-dimensional multicellular tumor spheroid culture. (**a**,**b**) MCTS morphology of (**a**) SiHa and (**b**) C33a tumor spheroids grown for seven days followed by 2-day treatment at the indicated concentrations of BAY-1143572, crinamine, cisplatin, FIT-039, carboplatin, 5-FU, gemcitabine, or DMSO. (**c**,**d**) Dose-response plots for (**c**) SiHa and (**d**) C33a tumor spheroids after treatment with various concentrations of different compounds. Cell viability was assayed and normalized to the DMSO control. IC_50_ values were calculated from two independent experiments performed in duplicate or triplicate and are presented as the mean ± SD.

**Figure 3 biomolecules-09-00494-f003:**
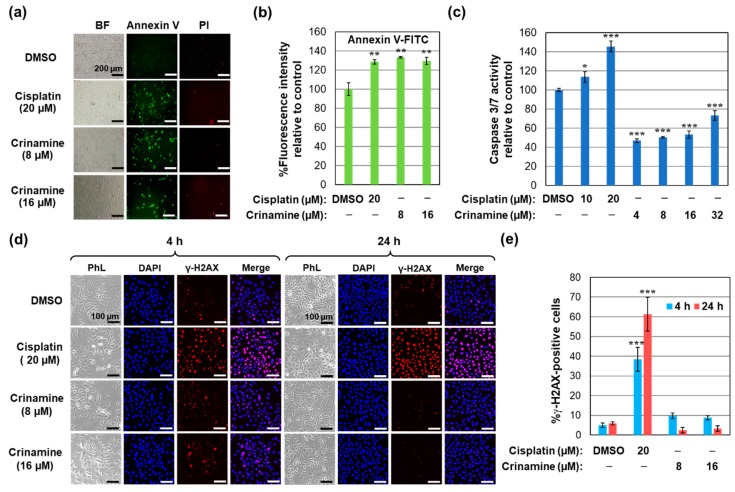
Crinamine-mediated apoptosis activation in cervical cancer cells. (**a**) Annexin V and PI staining of SiHa cells treated with cisplatin, crinamine, or DMSO for 16 h. (**b**) Fluorescent intensity of Annexin V staining from (**a**); results are shown as means ± SD from triplicate measurements. (**c**) Caspase-3/7 activity of crinamine in SiHa cells non-treated or treated with the indicated concentrations of cisplatin or crinamine for 24 h. Caspase activity is normalized to the DMSO control and is reported as means ± SD from triplicate experiments. (**d**) Immunofluorescence staining of histone γ-H2AX in SiHa cells treated with the indicated concentrations of cisplatin, crinamine, or DMSO for 4 and 24 h. Panels represent phase contrast (PhL) images, DAPI nuclear staining (blue), γ-H2AX foci (red), and merged images. (**e**) Percentage of γ-H2AX foci-positive cells quantified from five fields per treatment obtained from duplicate measurements. Results are representative of data observed on two separate occasions. **p* < 0.05; ***p* < 0.01; ****p* < 0.001.

**Figure 4 biomolecules-09-00494-f004:**
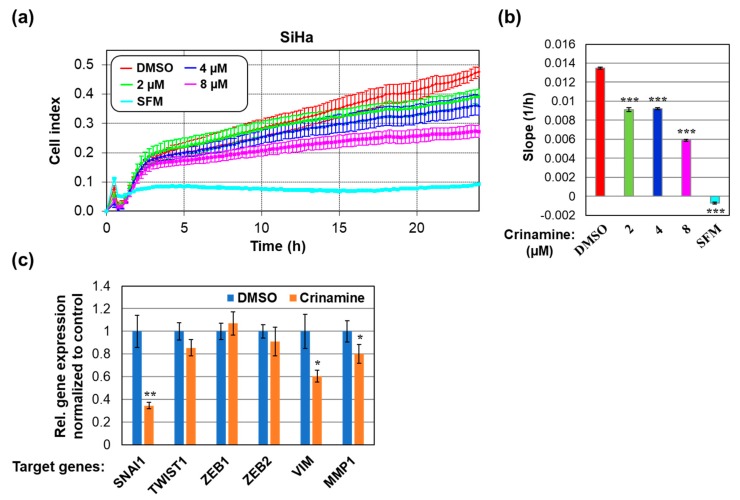
Inhibitory effects of crinamine on cervical cancer cell migration. (**a**,**b**) Real-time migration of SiHa cells treated with the indicated concentrations of crinamine or DMSO. The rate of cell migration is presented as (**a**) cell index values and (**b**) the slope of the line between the 5 h and 20 h time points. Data are shown as mean ± SD from triplicate measurements. The results presented here are representative of data observed in two independent experiments. (**c**) The effect of crinamine on mRNA expression of key genes involved in EMT and metastasis. SiHa cells were treated with DMSO or 12 µM crinamine for 8 h followed by gene expression analysis by RT-qPCR. RNA levels were normalized internally to U6 snRNA and RPS13 and externally to the DMSO control. Bars represent the mean ± SD from triplicate measurements. **p* < 0.05; ***p* < 0.01; ****p* < 0.001.

**Figure 5 biomolecules-09-00494-f005:**
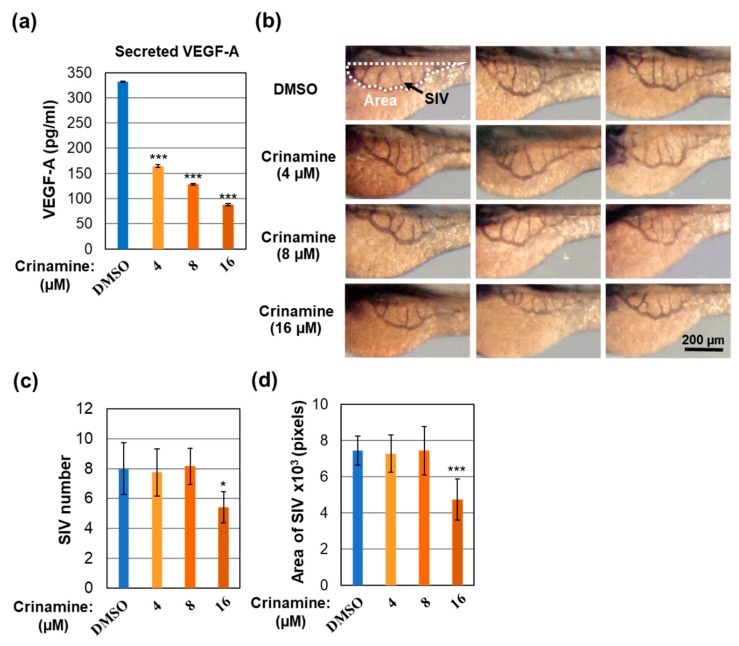
Crinamine-mediated inhibition of VEGF-A secretion and *in vivo* angiogenesis. (**a**) Effect of crinamine on VEGF-A release from SiHa cells. Cells were incubated for two days with the indicated concentrations of crinamine or DMSO. Secreted VEGF-A was quantified from cell culture medium by the Ella automated immunoassay platform and is reported as mean ± SD from triplicate measurements. (**b**–**d**) Effect of crinamine on SIV development in zebrafish embryos. (**b**) Representative alkaline phosphatase staining of three days post-fertilization zebrafish embryos treated with the indicated concentrations of crinamine or DMSO for three days. Images were obtained with a 3.2× objective. (**c**,**d**) Quantification of the number and area of SIVs from (**b**). Data are expressed as the mean ± SD (*n* = 12 per group). * *p* < 0.05; *** *p* < 0.001.

**Figure 6 biomolecules-09-00494-f006:**
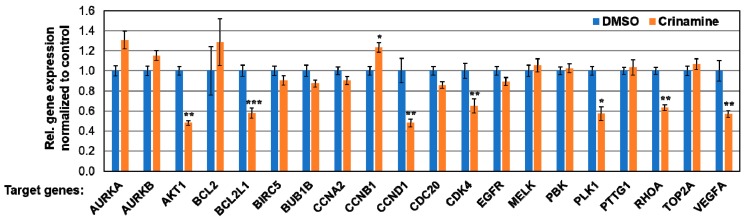
Effect of crinamine on mRNA expression of selected cervical cancer-related genes. SiHa cells were treated with DMSO or 12 µM crinamine for 8 h. Gene expression analysis was assessed by RT-qPCR. Expression levels were normalized internally to PGK1, RPS13, and U6 snRNA, and externally to the DMSO control. Bars indicate the mean ± SD from duplicate measurements. * *p* < 0.05; ** *p* < 0.01; *** *p* < 0.001.

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
