# Peer review of "Crinamine Induces Apoptosis and Inhibits Proliferation, Migration, and Angiogenesis in Cervical Cancer SiHa Cells"

_biomolecules, 2019, doi:10.3390/biom9090494_

Round 1
Reviewer 1 Report
Authors demonstrated the anti-proliferation, -migration and -angiogenesis of crinamine against cervical cancer cells. It is well designed and well written article. But some revisions should be done before publication.
Does crinamine have selective anti-cervical cancer effect? The drug failed to have cytotoxic effect on Hela and all the normal cell lines were influenced by the drug somehow at low doses too. Do they have statistical significance between crinamine-exposed normal cells and cervical cancer cells? Also, the title is about cervical cancer but C33a and Hela were less influenced by the crinamine. After figure 2, all the experiences were in SiHa only. Then howcome crinamine have anti-cervical cancer effect? It seems like the drug is effective only on SiHa cell line. Thus the title should be changed. Authors didn't show the direct mechanism of crinamine-induced apoptosis. Among decreased molecules such as AKT1, BCL2L1, CCND1, CDK4, PLK1, RHOA, which is the main target molecule of crinamine?
Author Response
"Please see the attachment."

Reviewer 2 Report
This manuscript (“Crinamine Induces Apoptosis and Inhibits Proliferation, Migration, and Angiogenesis in Cervical Cancer Cells”) has a clear scientific goal to study anti-cancer effects of a plant compound called crinamine isolated the bulbs of Crinum asiaticum. Overall, the manuscript is well-written, easy to follow, and based on the collection of comprehensive results, which were obtained using traditional methods of molecular and cellular biology except for less common but still valid biophysical technique of impedancemetry of cell cultures. Although the message of this manuscript is very clear, there are several points which require clarifications or revisions including the data presentation and statistics.
Lines 131-132:”…the number of viable cells was quantified using the CellTiter-Glo® 3D cell viability assay (Promega)”.
This assay does not allow to quantify the number of viable cells per se. This is a test for cell viability, i.e. the percentage of live cells.
Line 177: “Gene expression was normalized to at least two reference genes (U6, RPS13, or…”
This reviewer is not aware about a gene called U6. The authors most likely mean U6 snRNA. If it is true, it should be specified as required.
Lines 217-219: “To screen for potential anticancer activity, we first investigated the cytotoxicity of crinamine in normal and cervical cancer cell lines 218 using the MTS assay.”
The MTS assay was never mentioned in the Materials and Methods section. If the authors mean the CellTiter 96® AQueous Non-Radioactive Cell Proliferation Assay, they should specify it.
Line 233 and similar in the following text: “mean ± 1 standard deviation (SD)”
Do not need to add number 1 to this definition. It is obvious.
Figure 2a,b and Line 262: “As shown in Figure 2, crinamine strongly decreased tumor spheroid size…”
It is not clear from these images whether there were significant differences between tumor spheroid sizes because measurements are not reported.
295-297: “Whereas treatment of SiHa cells with 10 and 20 μM cisplatin for 24 h led to a significant increase in caspase-3/7 activity, treating cells with crinamine did not.”
This statement is based on the Figure 3, which actually shows that there is a significant decrease of caspase 3/7 activity. This finding however never mentioned or discussed.
Lines 308-310: “(e) Percentage of -H2AX foci-positive cells quantified from 5 fields per treatment obtained from duplicate measurements. Results are representative of data observed on two separate occasions. *, P < 0.05; **, P < 0.01; ***, P < 0.001.”
Statistics - How were P values calculated based on two separate occasions?
Subsection 3.4, Lines 348-349 and 372-374: “Treatment of SiHa cells with 2, 4, and 8 μM crinamine led to a significant decrease in cell migration in a dose-dependent manner (Figures 4a and 4b)” …..” As shown in Figure 4c, treating SiHa cells with 12 μM crinamine for 8 h significantly inhibited expressions of 373 SNAI1 and VIM…”
Why did the authors use significantly different concentrations of crinamine to study cell migration and expression of cell migration-related genes? It would be important to test cell migration in the presence of 12 μM crinamine.
Figure 6, Lines 433-434: “Bars indicate the mean ± 1 SD from duplicate measurements. *, P < 0.05; **; P < 0.01; 433 ***, P < 0.001.”
Again statistics, it is not clear how P values were calculated based on duplicate experiments.
Author Response
"Please see the attachment."

Reviewer 3 Report
The manuscript by Khumkhrong et al shows how a natural product like crinamine affects several stages of tumoral progression in cervical cancer cells.
The manuscript is clear and positions cervical cancer in similar scenarios as for other cancer types with respect the possible use of crinamine. Authors also provide some insights into the mechanisms of the antitumoral effects exerted by crinamine.
Some considerations should be taken into account prior to publication:
1) authors often refer to assays that have been normalized to DMSO, however it is not easy to identify which percentage of DMSO was used in each assay to be used in the normalization. Please correct this throughout the text.
2) I miss in the discussion a comparison between the effective concentrations the authors determine and the crinamine concentrations reported for other cancer cell types.
3) Page 6 line 243 "..every 30 min for 4 days. I rather use every 30 min during 4 days.
4) Page 7 line 268. FIT-039 and 5-FU in both SiHa and C33s cells. I believe it would more appropriate to use FIT-039 or 5-FU instead of "and".
5) Page 7 Line 272 figure legend. (a,b) cell morphology of....Strictly speaking it is not the cell morfology what is shown in the figure. It would rather be more MCTS morfology or size.
Author Response
"Please see the attachment."

Reviewer 4 Report
This paper described the anticancer effects of crinamine in both cancer cells and normal cells. The results showed it can selectively kill cancer cells over normal cells, and it exhibited potency in suppressing angiogenesis. It was an interesting compound and the paper was also well presented. As a reader I would like to know more:
If possible, please explain the reason for the choice of positive control (CDK9 inhibitors).
Why use the different concentrations of those drugs in the experiment of Tumor Spheroid Growth?
So far as I know, the authors applied different doses of crinamine in different experiments, such as 50, 100 μM in Figure 1c-1f (aren't they too high?); 8, 16 μM in Figure 3 but 12 μM in Figure 6.
In section 3.6, please explain why to test all these mRNAs and define all the abbreviations first.
Author Response
"Please see the attachment."
